# The Moderating Role of Sociodemographic Factors in the Relationship between Physical Activity and Subjective Well-Being in Chilean Children and Adolescents

**DOI:** 10.3390/ijerph182111190

**Published:** 2021-10-25

**Authors:** Sergio Fuentealba-Urra, Andrés Rubio-Rivera, Mònica González-Carrasco, Juan Carlos Oyanedel, Cristian Céspedes-Carreno

**Affiliations:** 1Facultad de Educación y Ciencias Sociales, Universidad Andres Bello, Concepción 4030000, Chile; juan.oyanedel@unab.cl; 2Facultad de Economía y Negocios, Universidad Andres Bello, Fernandez Concha 700, Las Condes, Santiago 7550000, Chile; andres.rubio@unab.cl; 3Facultad de Psicología, Universidad Diego Portales, Santiago 8320000, Chile; 4Research Institute on Quality of Life (IRQV), Universitat de Girona, 17004 Girona, Spain; monica.gonzalez@udg.edu; 5Facultad de Administración y Economía, Universidad de Santiago de Chile, Av. Libertador Bernardo O’Higgins 3363, Estación Central, Santiago 9170020, Chile; cristian.cespedes@usach.cl

**Keywords:** physical activity programs, school health, physical education, vulnerability, school children

## Abstract

Background: The relationship between physical activity habits and well-being is widely recognized; however, the interaction that these variables have with sociodemographic factors throughout life is only partially addressed in the literature, particularly in children and adolescents. The aim of this article is to analyze the moderating effect of sociodemographic factors and the possible interaction of these moderations in the relationship between physical activity and subjective well-being in children and adolescents. Methods: This cross-sectional study considered a sample of 9572 children and adolescents from 10 to 19 years of age, students of primary and secondary schools in all regions of Chile. Subjective well-being and physical activity habits were measured using self-report questionnaires. Socioeconomic level was established from the school vulnerability index (SVI) of each student’s school. Results: Simple moderation analyses revealed that the higher the age and the lower the SVI, the stronger the relationship between physical activity habits and subjective well-being. From a double moderation analysis, it could be observed that the age of the subjects is the most relevant moderator in the relationship between physical activity habits and perceived well-being in young people. Conclusions: This study highlights the importance of considering these factors and their interaction when generating programs or public policies to improve physical activity habits and well-being in children and adolescents.

## 1. Introduction

Physical activity (PA) can be defined as a set of behaviors initiated by body movement, generated from the process of the voluntary contraction of skeletal muscles [1]. It corresponds to a muscular action, which involves energy expenditure and is related to the development of actions of daily living, exercise, work and sports practice [2]. The World Health Organization [3] has developed a series of recommendations regarding its practice for each stage of the life cycle. These guidelines suggest that children and adolescents should perform at least 60 min of physical activity daily. Despite the above, research shows that it is precisely in the transition from childhood to adolescence that the practice of physical activity decreases considerably [4,5,6].

The transition from childhood to adolescence is recognized as a decisive stage in human development, during which individuals may experience intense physical, psychological and emotional changes [7]. This stage is crucial because it is when autonomy increases with respect to the development of healthy lifestyle habits [8,9]. Well-being at this stage is related to the practice of games, movement, sports, outdoor activities, exercise and physical education classes [10,11,12]. Well-being corresponds to a condition that, from a hedonic perspective, is understood as people’s satisfaction with different areas of their life and their life in global terms. This is recognized as subjective well-being (SW) and is related to affect and, in particular, to the positive or negative emotional experiences that people have. In general terms, SWB is based on cognitive and affective processes, associated with perceptual parameters inherent to each individual [13,14]. Physical activity involves processes of the same nature [7,15], from which point its relationship with perceived well-being arises. In this regard, research indicates that physically active adolescents have a higher level of satisfaction with life [16,17]. It has been observed that even minimal amounts of PA can be positive for well-being, cancelling out the negative influence of factors such as being overweight [12,18]. At the neurophysiological level, it has been established that the practice of physical activity could improve the perception of well-being through effects such as the increase in the release of endorphins, in the expression of brain-derived neurotrophic factor and the formation of new capillaries that, together, improve the structural and functional conditions of the nervous system [19].

The theory of affective regulation emerges within the explanatory models. This suggests that the continuous practice of PA produces improvements in mood and a decrease in anxiety, irritability and stress, modulating and regulating psychological and neurobiological processes that generally act positively on the well-being of children and adolescents [19]. The practice of physical activity is associated with emotional regulation skills, including self-regulation, which plays an important role in the consolidation of healthy lifestyle habits [7,8], and which in turn are associated with higher levels of well-being and quality of life from childhood [20,21]. On the contrary, sedentary habits are associated with poor mental health [22] in emotional terms and with an increase in the presence of maladaptive coping patterns, associated with a reduction in the use of emotional regulation strategies. These aspects are related to low levels of well-being in children and adolescents [7,23,24]. Physical activity can have a positive impact on the mental health of adolescents [25]; this aspect is relevant given that mental health problems are increasingly frequent in children and adolescents [26] and are not properly addressed [27,28], which can have serious consequences in adult life [25,29]. In this line, the theory of self-determination corresponds to an approach towards the motivation of the human being centered on the resources necessary for the development of personality and the self-regulation of behavior [30,31]. It establishes that the motivation can be extrinsic or intrinsic in nature. The latter is associated from childhood with factors such as inherent satisfaction, spontaneous interest, the challenge of assimilating and mastering as well as exercising exploration and learning abilities. The transfer of an external to an internal motivation is a critical process that can be substantially modified in the transition from childhood to adolescence and is associated with an increase in autonomy that eventually promotes the development of self-determined actions [32]. Finally, when the practice of physical activity responds rather to the adolescent’s own preferences and needs, it is then positively associated with SWB [33]. The practice of PA as a habit in the long term is associated with internal motivations related to feeling competent, having autonomy and relating to others. These are elementary psychological needs that facilitate personal and social development, self-regulation and well-being [19,31].

On the other hand, age, gender and socioeconomic status are determining sociodemographic factors, considered for decades in the study of SWB [13,27,34,35] and the practice of PA [36]. In Chile, investigations that have examined subjective well-being [37] and physical activity [38] in children and adolescents also account for their consideration in the analyses. Despite the above, it is observed that in a large part of the investigations that analyze the relationship between physical activity and well-being in children and adolescents, they consider only parts of these factors, also showing results at a descriptive level (for example when characterizing study samples) or in analyses that do not consider well-being or physical activity. This is reported, for example, in reviews [39,40,41]. In this context, moderation analyses emerge, from a statistical perspective [42], as an adequate alternative that clearly identify how gender, age and socioeconomic status can modify the relationship between physical activity habits and SWB. In this sense, the moderating effect of sociodemographic factors and the possible interaction of these moderations could broaden the understanding of the relationship between physical activity habits and well-being and their determining factors in children and adolescents [43].

Considering the evidence presented above, analyzing the moderating effects of sociodemographic factors and the possible interactions of these moderations, in the relationship between physical activity and subjective well-being could also be useful for the following purposes: first, broadening the knowledge regarding the underlying mechanisms in the relationship between physical activity habits and the well-being of children and adolescents; second, it would make it possible to establish more precise guidelines when designing programs and public policies on the well-being of children and adolescents generated from the practice of physical activity; and third, analyzing the moderating effect of gender, age and socioeconomic status in Chile could be of particular relevance given that it is a developing country with still-high levels of inequality [44,45]. We hypothesize that sex, age and socioeconomic level have a moderating effect on the relationship between physical activity habits and subjective well-being and that these moderations also interact with each other in the relationship between PA and SWB in Chilean children and adolescents.

## 2. Materials and Methods

This research corresponds to a non-experimental, cross-sectional study with a descriptive-correlational scope.

### 2.1. Participants

The sample consisted of 9572 male (4650; 48.6%) and female (4922; 51.4%) children and adolescents between 10 and 19 years of age (M = 13.88, SD = 2.08), students from 5th year of primary school to 4th year of secondary school, from different types of educational establishments (public, subsidized and private) from all regions of Chile. The sampling was probabilistic, two-stage and stratified (considering the dependence of the schools and their vulnerability index). The first probabilistic sampling unit was the school and the second one was the class chosen from each grade (in Chile, each grade has more than one class, e.g.,: 6th A, 6th B, 6th C). The official list of educational establishments 2017 of the Chilean Ministry of Education was used as the sampling frame.

### 2.2. Data Collection Method

Schools randomly selected to be part of the study were contacted. The scale was applied through self-report questionnaires during the 2017 school year, during regular school hours, as part of an instrument that included other scales in a larger study. A previously trained group, who carried out the process under standardized conditions, developed the measurements. The students were consulted in the permanent presence of a teacher, completing the questionnaires in an optimal classroom environment, with the help of researchers.

### 2.3. Instruments

#### 2.3.1. Subjective Well-Being (SWB)

To examine subjective well-being in adolescents, the 7-item Personal Wellbeing Index in its school version (PWI-SC) was used [46]. The psychometric properties of this scale were previously analyzed in Chile, establishing its reliability and validity [47]. The scale has 11 levels; from 0, which means totally dissatisfied, to 10, which means totally satisfied, and various areas of life were considered: standard of living, personal health, achievement in life, personal relationships, personal safety, feeling part of the community and future security. It considered questions such as, “Please tell us to what extent you are satisfied with what can happen later in your life”. In this study, the reliability of the scale score, given by its Cronbach’s Alpha, which considered the 7 items of the scale, was 0.86.

#### 2.3.2. Habits of Physical Activity and Inactivity (PA)

The Eating and Physical Activity Habits Questionnaire for schoolchildren developed and validated by Guerrero et al. [48], who reported good psychometric properties for the instrument (validity and reliability), was used. This instrument includes two dimensions with a total of 27 items, of which 18 are directed to know the eating and nutrition habits and 9 are directed to know the physical activity and inactivity habits of the subjects. Due to the extension of the scale, it had to be reduced, leaving 7 items specifically referring to physical activity and inactivity habits, that cover fundamental aspects such as the type and weekly frequency of their practice [3]. The scale has 5 frequency levels, from 1, which means never, or less than one time per month, to 5, which means daily. It included questions such as, for example, “Please indicate the frequency with which you perform the following activities: I do physical activities and/or sports with my family”. In this study, the reliability of the physical activity/inactivity habits part of the scale score (7 items), given by its Cronbach’s Alpha, which considered the 7 items of the scale, was 0.71. Furthermore, a CFA analysis was carried out to analyze the construct validity of this dimension, yielding acceptable fit indexes. (χ^2^(14) = 3395.92, *p*-value < 0.001; RMSEA = 0.00; CFI = 0.71 and SRMR = 0.09).

#### 2.3.3. Socioeconomic Level (SEL), Gender and Age

To establish the SEL of the participants, our study used the school vulnerability index (SVI), which is established for each educational establishment based on characteristics such as family income, housing, number of members and other social characteristics, such as the educational level of the parents, information contained in the civil registry, the national health fund (FONASA) and the social protection card. The SVI corresponds to a categorization (low, medium or high) according to the percentage of students in the establishment that qualify as vulnerable according to the criteria of the National Board of School Aid and Scholarships (JUNAEB). This measure is considered an indirect but reliable way of measuring SEL [37]. In our study, students who belonged to an establishment with high SVI (SVI = 1) were considered vulnerable and students who belonged to an establishment with medium or low SVI were considered non-vulnerable (SVI = 0). To establish gender, two options were given to respond with (0 = boys; 1 = girls), and to measure age, participants were asked how old they were (in years) at the time of answering the questionnaire. Both gender and SVI were considered as dichotomous categorical variables, while age was considered as a continuous variable.

### 2.4. Statistical Analysis

First, descriptive analyses were performed (percentages, mean and standard deviation for each of the variables of interest). Comparisons between PA and SWB scores were made for the groups by gender, age (10–12, 13–15 and 16–19 years; reference age for the last levels of primary, first-second level of secondary and third-fourth level of secondary school, respectively) as well as school vulnerability, using the Student’s T test and one-way ANOVA, after demonstrating the assumptions of normality (Kolmogorov–Smirnov) and homoscedasticity (Levene Test). Additionally, post hoc analyses were considered for multiple comparisons. The effect size was analyzed following the recommendations of Cohen [49]. The moderating role of gender, age and socioeconomic level in the relationship between PA habits and SWB was examined using a simple moderation analysis (for each of the moderators). Subsequently, a multiple moderation analysis was performed simultaneously considering all the moderators that were statistically significant in the simple moderation analyses with the aim of analyzing which had greater strength as a moderator in the relationship between PA and SWB. All analyses were performed with IBM-SPSS Version 25 software (IBM Inc., Armonk, NY, USA) and its PROCESS tool [42].

## 3. Results

### 3.1. Descriptive Results

Table 1 shows frequencies, averages and standard deviation of the group for each of the variables considered in the research.

Table 2 shows the scores, considering the total score obtained by the instruments used to measure PA and SWB habits, according to gender, age groups, years in the final primary education process, and vulnerability condition. Significant differences and small effect sizes are observed in the level of physical activity and subjective well-being when considering gender as a comparison factor. Men in the sample achieved higher scores compared to women. When considering age range as a comparison factor, the results show significant differences. Medium effect sizes are also observed. The post hoc analysis reveals that differences are present among the three groups. Older adolescents (16–19 years old) are less physically active and have a lower perception of well-being than their peers aged 13–15 years, who in turn are less active and have a lower perception of well-being than their peers aged 10–12 years. Finally, there are no significant differences in the level of physical activity of adolescents considering their SVI, but there are differences between these groups when comparing perceived well-being. Individuals categorized as socioeconomically vulnerable have lower well-being than their non-vulnerable peers.

### 3.2. Moderation Analysis

This section presents the results corresponding to the single and double moderation analyses based on sociodemographic factors for the relationship between physical activity and subjective well-being.

#### 3.2.1. Simple Moderation Analysis

These models considered PA as the independent variable, SWB as the dependent variable, and gender, age and vulnerability status as moderating variables. The mean, low and high values of the moderating variables were established from their mean plus/minus one standard deviation. For the three models, a confidence interval based on the bootstrapping method (5000 resamples) was considered to determine the significance of the moderating effect.

When analyzing the case of gender, it can be observed descriptively (see Figure 1) that the effect of PA on SWB is the same for men and women (b = 0.54). As we can see in Table 3, the moderation analysis showed that the interaction between gender and PA was not statistically significant (*p* = 0.87) in predicting SWB; gender in this case did not have a moderating effect on the relationship between PA and SWB.

In the analysis of age, it can be observed descriptively (see Figure 2) that as age increases, the effect of PA on SWB also increases. In this model, age was considered as a continuous variable, with the mean age level being the mean age of the sample and the low and high levels calculated from the addition/subtraction of an SD from the mean. As we can see in Table 4, the moderation analysis showed that the interaction between age and PA was statistically significant (*p* < 0.001) in predicting SWB. So, the moderating effect of age on the relationship between PA and SWB is statistically significant. The difference in the association level of the relationship between PA and SWB is as high as 33.33%, comparing the low and high age levels. Additionally, the change in R^2^ value when including the interaction component was 0.16%.

When analyzing the socioeconomic level by means of the categorized SVI, it can be observed descriptively (see Figure 3) that the effect of PA on SWB is 1.25 times greater for the case of students without socioeconomic vulnerability compared to students in vulnerable situations. As we can see in Table 5, the moderation analysis showed that the interaction between socioeconomic vulnerability and PA was statistically significant (*p* < 0.01) in predicting SWB. Vulnerability has a moderating effect on the relationship between PA and SWB. The difference in the association level of the relationship between PA and SWB is as high as 20.00%, comparing vulnerable and not vulnerable conditions. Additionally, the change in R^2^ value when including the interaction component was 0.10%.

#### 3.2.2. Double Moderation Analysis

Since moderation was statistically significant, for the models that included age and socioeconomic vulnerability status, the interaction of these moderators was analyzed in a double moderation model (See Figure 4). As we can see in Table 6, the results showed that by including these moderating variables together, both interacted statistically significantly with PA when predicting SWB (B = −0.10, *p* < 0.05 for the case of vulnerability, and B = 0.04, *p* < 0.001 for the case of age). That is, when including these variables in the model together, both have a statistically significant moderating effect, with the moderating effect of socioeconomic vulnerability being 2.5 times smaller than the moderating effect of age. Table 7 shows how, as age increases and there is no vulnerability, the effect of PA on SWB is greater. Additionally, the change in R^2^ value when including the interaction component was 0.25%.

## 4. Discussion

This study analyzes the moderating effect of sociodemographic factors and the possible interaction of these moderations in the relationship between physical activity habits and subjective well-being of Chilean children and adolescents.

The results revealed that the relationship between physical activity habits and subjective well-being becomes stronger in older subjects (16–19 years); age then has a moderating effect on the relationship between both variables. These results are in line with the findings of review and meta-analysis studies, such as the one developed by Rodríguez-Ayllon et al. [41]. This study examined a set of descriptive and experimental investigations and of the latter, the analyses revealed that the practice of physical activity generated a significant effect on the mental well-being of children and adolescents. However, when the analysis was stratified by age group, it turned out that this effect was significantly appreciated only in the adolescent group. In contrast to the above, descriptive research carried out independently for children [50] and adolescents [17,51] show that the habitual practice of physical activity is independently and consistently related to higher levels of perceived well-being. In this sense, Chatzisarantis et al. [52] point out that the practice of PA is promoted to a greater extent when it is perceived as an autonomous initiative generated from an intrinsic motivation of the individual. Such motivation and the internalization of self-regulatory behaviors generate conditions for greater autonomy and the development of self-determined actions [32]. In this sense, the practice of physical activity in children and adolescents is part of the essential actions that increase autonomy in their development, which is an aspect that regulates a part of healthy behaviors [31] and is finally related to a higher perception of well-being [53].

Our study also shows that the socioeconomic level, expressed in terms of school vulnerability, is also a determining factor. The relationship between physical activity habits and subjective well-being becomes stronger in “non-vulnerable” subjects. One of the difficulties in contrasting with the literature is the variety of socioeconomic conditions described in the research, which makes it difficult to establish their role with precision [43]. Despite the above, Booker et al. [6] point out that higher levels of well-being may be explained in non-vulnerable populations by the interaction of related factors such as the level of education of the parents, living conditions and the environment [54]. Therefore, the relationship between well-being and physical activity will be stronger if there is a positive and reinforcing social context for its practice [16]. This is related to the conditions of autonomy and support, structure, and involvement described by Decci and Ryan, [55] which indicate that they should be first favored by parents and teachers, in terms of support for social conditions that favor the autonomy of young people. Studies carried out in Chilean children and adolescents confirm that the practice of physical activity is strongly associated with their socioeconomic level. Children and adolescents from families with higher economic incomes perform a greater amount of physical activity, reaching the international recommendations [3] regarding its practice [56] to a greater extent. For Chilean children and adolescents, the gap in physical activity habits is largely explained by existing socioeconomic inequality. This is mainly reflected in aspects such as access and quality of sports infrastructure, the promotion of physical activity within schools and the allocation of resources for its development [57]. Regarding subjective well-being, it is important to highlight that, in developed countries with lower levels of social and economic inequality, the relationship between SWB and SES is practically non-existent [58]. Despite the fact that Chile in the 30th Global Human Development Report, presented by the United Nations Development Program [44] reaches the first place in Latin America and the 43rd position among 189 countries, when the Human Development Index is adjusted by inequalities, it falls 11 places in the world ranking. The report indicates that inequalities are mainly marked by issues associated with gender and income level. Research carried out in Chile in children and adolescents generally shows high levels of SWB, independent of socioeconomic conditions [59]. In this regard, Oyanedel et al., 2015 [37] warn that this phenomenon can be attributed to the so-called “vital optimism bias”, which implies that very young subjects tend to report high levels of satisfaction. They also point out that the results, particularly in adolescents, should differ given that at that age they are more critical regarding their vital conditions.

Finally, a significant interaction effect is observed among the moderations described above. The results show that the relationship between PA and SWB is stronger in non-vulnerable adolescents (16 to 19 years old). This result reinforces what has been described above, and is in line with the literature that precisely suggests that sociodemographic factors should be considered within the analysis of moderation effects and also as a whole for the analysis of the relationship between PA and SWB in children and adolescents [60].

### 4.1. Strengths

This study’s strengths have to do with the representativeness of the sample in terms of size and sociodemographic diversity, in addition to the use of validated instruments and standardized methods for measurement.

### 4.2. Limitations

Some limitations are recognized within the research. First, the results are based on subjective self-reporting; in this sense, it is assumed that the results may be biased by conditions of social desirability of the young people surveyed. Secondly, in our study it was not possible to consider the level of maturity of the subjects. The socioeconomic level was reported from the vulnerability index, which is a measure that summarizes the socioeconomic conditions of all students and, therefore, may not reflect the particular reality of each of them. Thirdly, the study had a cross-sectional design, so it is beyond its scope to show evidence on the predictive role of PA on SWB; therefore, the association between these variables was proposed under the theoretical assumption of causality. Future research should consider these limitations in order to obtain even more enlightening results.

## 5. Conclusions

Our study has shown that sociodemographic factors play a moderating role in the relationship between physical activity and subjective well-being in children and adolescents. The results are consistent with the considerations of the World Health Organization, which warns of the importance of these factors in the design of its guidelines [3]. The study has shown that the moderations made for age and socio-economic level expressed in terms of vulnerability interact and significantly moderate the relationship between physical activity habits and subjective well-being. In specifying the above, it can be seen that age had a greater moderating effect. Finally, it can be pointed out that the results obtained in this study allow us to broaden the understanding of the relationship between healthy lifestyle habits and the well-being of children and adolescents, establishing underlying mechanisms that interact in these phenomena. It is proposed to consider these and other findings when establishing guidelines for the design of physical activity programs aimed at improving well-being in children and adolescents.

## Figures and Tables

**Figure 1 ijerph-18-11190-f001:**
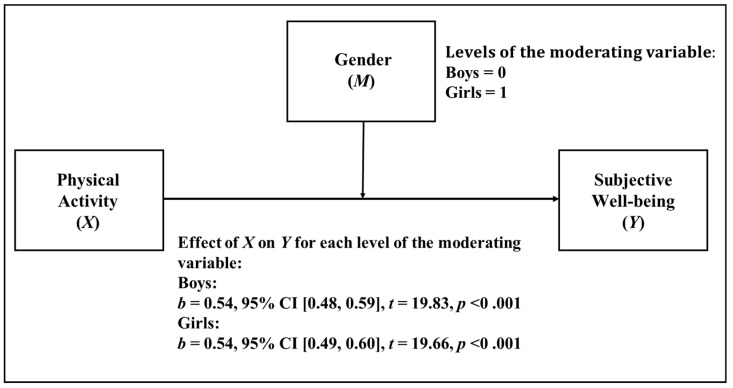
Simple moderation model in which gender is considered as moderator.

**Figure 2 ijerph-18-11190-f002:**
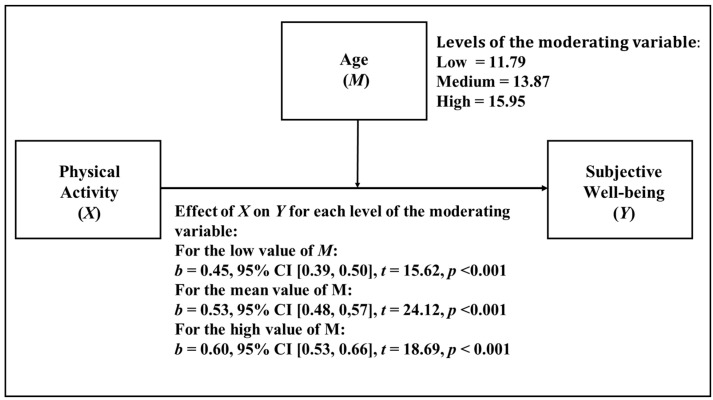
Simple moderation model in which age is considered as moderator.

**Figure 3 ijerph-18-11190-f003:**
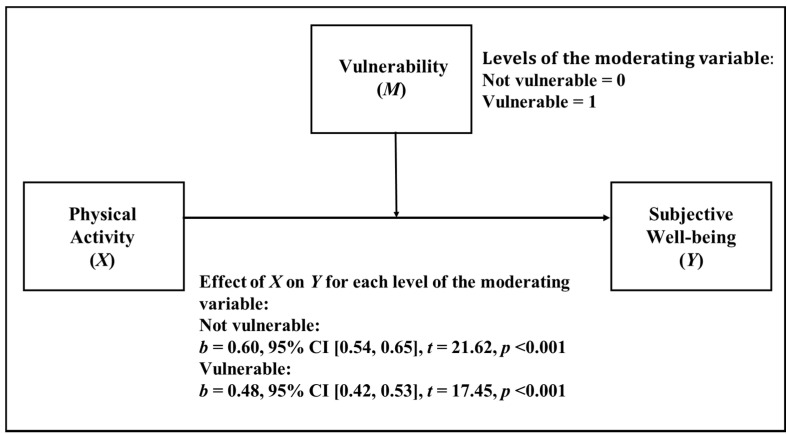
Simple moderation model in which socioeconomic vulnerability is considered as moderator.

**Figure 4 ijerph-18-11190-f004:**
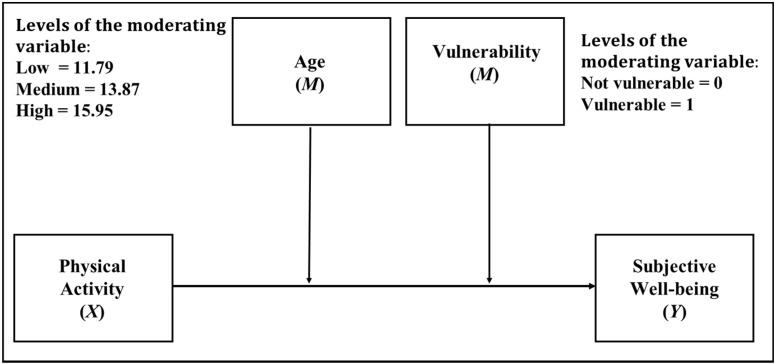
Double moderation model in which age and socioeconomic vulnerability are considered as moderators.

**Table 1 ijerph-18-11190-t001:** Participants’ Characteristics (*n* = 9572).

Girls, (*n*, %)	4922 (51.4)
School Vulnerability (*n*, %)	4634 (48.3%)
Age, years	13.88 (2.08)
Physical Activity	2.84 (1.00)
SWB (PWI-SC)	8.25 (1.65)

Date are mean (SD) or number and proportions (%); Subjective well-being by personal well-being index.

**Table 2 ijerph-18-11190-t002:** Total scores for physical activity habits and subjective well-being.

Physical Activity	*n*	Mean (SD)	SE
Boys	4650	3.07 (0.99)	0.46
Girls	4922	2.61(0.96) **	
10–12 years old	2894	3.22 (1.03) ^b,c^	0.28
13–15 years old	4281	2.74 (0.96) ^a,c^	
16–19 years old	2397	2.49 (0.88) **^,a,b^	
Non vulnerable	4938	2.85 (0.98)	0.03
Vulnerable	4634	2.82 (1.02)	
**SWB (PWI-SC)**			
Boys	4650	8.31 (1.63)	0.07
Girls	4922	8,19 (1.66) **	
10–12 years old	2894	8.64 (1.38) ^b,c^	0.15
13–15 years old	4281	8.19 (1.64) ^a,c^	
16–19 years old	2397	8.04 (1.68) **^,a,b^	
Non vulnerable	4938	8.33 (1.58)	0.09
Vulnerable	4634	8.17 (1.69) **	

Date are *n*, mean (SD). SWB (PWI-SC) = Subjective Well-Being by Personal Well-Being Index. ** Significant differences for the comparison of means of independent groups (*p* < 0.001). ^a,b,c^ post hoc significant (*p* < 0.001). SE, size effect.

**Table 3 ijerph-18-11190-t003:** Regression analysis of SWB on PA, considering the moderating effect of gender.

	b	se	t	*p*	LLCI	ULCI
constant	6.67	0.09	77.75	<0.001	6.50	6.84
gender	−0.01	0.12	−0.05	0.96	−0.24	0.22
PA	0.54	0.03	19.83	<0.001	0.48	0.59
interaction	0.01	0.04	0.16	0.87	−0.07	0.08

SWB: Subjective Well-Being. PA: Physical Activity. LLCI: Lower Limit of the Confidence Interval. ULCI: Upper Limit of the Confidence Interval.

**Table 4 ijerph-18-11190-t004:** Regression analysis of SWB on PA, considering the moderating effect of age.

	b	se	t	*p*	LLCI	ULCI
constant	9.24	0.44	20.92	<0.001	8.38	10.11
age	−0.18	0.03	−5.67	<0.001	−0.24	−0.12
PA	0.02	0.14	0.11	0.91	−0.26	0.29
interaction	0.04	0.01	3.57	<0.001	0.02	0.06

SWB: Subjective Well-Being. PA: Physical Activity. LLCI: Lower Limit of the Confidence Interval. ULCI: Upper Limit of the Confidence Interval.

**Table 5 ijerph-18-11190-t005:** Regression analysis of SWB on PA, considering the moderating effect of vulnerability.

	b	se	t	*p*	LLCI	ULCI
constant	6.57	0.08	77.37	<0.001	6.40	6.73
vulnerability	0.22	0.12	1.82	0.07	−0.02	0.45
PA	0.60	0.03	21.62	<0.001	0.55	0.65
interaction	−0.12	0.04	−3.19	<0.01	−0.20	−0.05

SWB: Subjective Well-Being. PA: Physical Activity. LLCI: Lower Limit of the Confidence Interval. ULCI: Upper Limit of the Confidence Interval.

**Table 6 ijerph-18-11190-t006:** Regression analysis of SWB on PA, considering the moderating effect of age and vulnerability.

	b	se	t	*p*	LLCI	ULCI
constant	9.24	0.45	20.50	<0.001	8.36	10.12
vulnerability	0.23	0.13	1.80	0.07	−0.02	0.49
PA	0.01	0.14	0.10	0.92	−0.27	0.30
age	−0.19	0.03	−5.83	<0.001	−0.25	−0.12
PA × vulnerability	−0.10	0.04	−2.40	<0.05	−0.19	−0.02
PA × age	0.04	0.01	3.88	<0.001	0.02	0.06

SWB: Subjective Well-Being. PA: Physical Activity. LLCI: Lower Limit of the Confidence Interval. ULCI: Upper Limit of the Confidence Interval. PA × vulnerability = Effect of the interaction between PA and vulnerability on SWB. PA × age = Effect of the interaction between PA and age on SWB.

**Table 7 ijerph-18-11190-t007:** Conditional effect of physical activity on subjective well-being, moderated by age and socioeconomic vulnerability.

Age	SV	Effect	SD	t	*p*	LLCI	ULCI
11.79	0	0.49	0.035	14.18	<0.001	0.43	0.56
11.79	1	0.39	0.038	10.44	<0.001	0.32	0.47
13.87	0	0.58	0.030	19.03	<0.001	0.52	0.64
13.87	1	0.48	0.031	15.28	<0.001	0.42	0.54
15.95	0	0.66	0.039	16.66	<0.001	0.59	0.74
15.95	1	0.56	0.038	14.55	<0.001	0.49	0.64

SV: School Vulnerability (0 = Not vulnerable; 1 = Vulnerable). LLCI: Lower Limit of the Confidence Interval. ULCI: Upper Limit of the Confidence Interval.

## Data Availability

The underlying research materials related to this paper are available from the corresponding author upon request.

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
