# Peer review of "The Moderating Role of Sociodemographic Factors in the Relationship between Physical Activity and Subjective Well-Being in Chilean Children and Adolescents"

_ijerph, 2021, doi:10.3390/ijerph182111190_

Round 1

Reviewer 1 Report

The authors should address the following before resubmitting the article:

Introduction

  • The introduction concludes without any testable hypotheses or clearly articulated research questions. The work is left to the reader to guess what the authors are going to test as part of the study. The authors should provide a rationale for the current study, which lead to testable hypotheses or research questions, as a final paragraph of their introduction. This is a major limitation of the Introduction component as it stands.
    • Further to this point, the importance of the current research isn't clearly explained to the reader. Following the literature reviewed in the last paragraph of this section, the reader doesn't have a clear picture what the current study can clarify / offer, and why this matters. The 'so what?' question the reader is left with throughout the Introduction is not answered sufficiently by the authors based on how the Introduction is currently presented, and this would benefit from further revision.
  • A clear rationale for the choice of frameworks or models that would predict relationships between PA and SWB is limited. For example, self-determination theory is mentioned in passing, but the reader must have existing knowledge to guess how the autonomous/controlling forms of motivation in SDT may be relevant to PA practice, but this is not clearly articulated to the reader. The theoretical / framework components of the Introduction are therefore not well articulated to the reader, making it difficult to understand the evidence basis that supports the research without prior knowledge. This is a major limitation of the Introduction.

Method

  • The reader does not know what the authors refer to in terms of course choices when describing the stratification method in the sentence  "The first probabilistic sampling unit was the school and the second was the course to be chosen within each level". What are the levels/course? This is not addressed again in the article, adding further confusion.
    • Given that the data is stratified, should the analyses have been conducted in a manner that integrates the complex survey structure?
  • Although the sample is large, clarifying that it meets adequate statistical power in the participants section would be valuable.
  • In the instruments section, clarify to the reader that scale scores are being referred to, and not the scales themselves, when describing reliability. 
  • Further justification of the adequacy of the PA measure in 2.3.2 is required given the modification of an existing measure's items without construct validity evidence presented.
  • In section 2.4, please clarify whether the PROCESS macro in SPSS centers the variables used to create the interaction terms. The potential for estimation bias due to strong collinearity between the base variables and their interaction terms is a core methodological consideration for moderation analyses.
  • Section 2.4 doesn't describe the analyses used for mean score differences presented in Table 2, nor does it explain why Age was split into arbitrary categories of 10-12 years old etc.

Results

  • In Table 2, statistical significance is presented as meaningful for mean score comparisons across the levels of the IVs, but realistically with a sample of ~9000 participants, even marginally-different comparisons will be significant. Present the effect sizes of these mean difference comparisons so the reader can understand practical significance. See the ASA's statement at https://www.amstat.org/asa/files/pdfs/p-valuestatement.pdf for further guidance on the use and interpretation of p values.
  • Regression tables to present the results, instead of using statistical sentences in a conceptual figure of a moderation model, would be more appropriate when presenting the moderation findings. Currently information such as the intercept value are missing, as are estimates of effect size reported in-text such as an R2 value, and change in R2 value, following the introduction of the interaction component to understand the change in variance it is associated with. Statements such as "The difference in the variance of the relationship between PA and SWB is as high as 1332%." on page 7 are effectively meaningless without knowing what the R2 values are before/after introduction of the interaction term, as the reader is unable to judge the size of the effect prior to the introduction of the interaction term.
  • In Figure 2, it is still unclear whether age has been entered as a continuous variable and the simple slope analysis is based on +- 1 SD as mentioned earlier in the results, or whether it has been evaluated as three nominal levels. Presenting a plot of the regression slopes varying by +- 1 SD and the mean, would be more effective compared to presenting a Figure containing statistical sentences here, and the former is more of the standard method of presenting moderation findings in sources such as Hayes' explanation of the PROCESS macro and how to employ it.
  • Section 3.2.2. would benefit from presenting a table of the regression results as noted earlier, in order to assist the reader with understanding how the model has been parameterised and the coefficients associated with each parameter.
  • Effect sizes are absent throughout the Results, which is a major limitation of this section. The reader is unable to judge the practical significance of any of the presented findings unfortunately.

Discussion

  • Starting the discussion, the authors note "In this sense, the aim of the present investigation was to analyze the moderating effect and possible interactions of the variables gender, age and socioeconomic level in the relationship between physical activity and subjective well-being in Chilean children and adolescents". Interactions between gender, age, and socioeconomic level were not clearly examined; the conceptual model in Figure 4 for example presents separate moderating effects of PA*Age and PA*Vulnerability, yet Table 3 presents PA*Age*Vulnerability findings. Reconciling the opening description of the study's aims against what has been presented in the Results, without the requested regression tables showing the coefficients of the model parameters, is challenging for the reader and reflects a major obstacle for clarity.
  • The literature reviewed shortly after the previous point revisits the content from the Introduction, but requires more emphasis on describing how the current findings are similar/vary from previous research, without repeating the Introduction's content again.
  • Much of the literature described in the second paragraph of the section subtitled 'Moderating effect of gender' does not seem to have a clear bearing on the moderating effect of gender. It instead broadly describes PA and wellbeing for reasons that are not clear.
  • A major limitation of the Discussion is tied to the coherency of the points being presented and the clarity of the writing. This section does not seem to have received the same care and editing as earlier sections, making it difficult for the reader. Consider the section 4.1 Strengths. Sentences feature missing words or content, such that by the time the reader reaches "Future research should consider these limitations..." (what limitations? Strengths are being described here?), it is clear that this section requires a significant revision for clarity.
  • It is difficult in the conclusion for the reader to understand the importance of the study, and how its findings are novel, based on the preceding Discussion content. The authors should try to clearly articulate the novelty of their findings and how they differ from other similar works in the literature.

Author Response

October 10, 2021

Dear reviewer

We appreciate each of your observations, they have generated a learning process and have undoubtedly allowed us to improve the quality of the document.

Below, we detail each of the improvements and comments generated from your observations. To facilitate your understanding, we have incorporated them into this document in the same order that you established.

We attach a new version of the document, highlighting the improvements in yellow. We hope in this opportunity to live up to your expectations.

Kind regards,

Sergio Fuentealba Urra

Professor

----------------------------------------------------------------------------------------

Introduction

  • The introduction concludes without any testable hypotheses or clearly articulated research questions. The work is left to the reader to guess what the authors are going to test as part of the study. The authors should provide a rationale for the current study, which lead to testable hypotheses or research questions, as a final paragraph of their introduction. This is a major limitation of the Introduction component as it stands.

Reply: we have incorporated in the introduction the hypotheses and the justification necessary for the investigation.

  • Further to this point, the importance of the current research isn't clearly explained to the reader. Following the literature reviewed in the last paragraph of this section, the reader doesn't have a clear picture what the current study can clarify / offer, and why this matters. The 'so what?' question the reader is left with throughout the Introduction is not answered sufficiently by the authors based on how the Introduction is currently presented, and this would benefit from further revision.

Reply: we have incorporated in the introduction some elements that allow us to understand what the study offers and clarifies.

  • A clear rationale for the choice of frameworks or models that would predict relationships between PA and SWB is limited. For example, self-determination theory is mentioned in passing, but the reader must have existing knowledge to guess how the autonomous/controlling forms of motivation in SDT may be relevant to PA practice, but this is not clearly articulated to the reader. The theoretical / framework components of the Introduction are therefore not well articulated to the reader, making it difficult to understand the evidence basis that supports the research without prior knowledge. This is a major limitation of the Introduction.

Reply: we have improved the articulation in the introduction for those aspects referring to the frameworks-models that predict the relationship between PA-SWB.

Method

  • The reader does not know what the authors refer to in terms of course choices when describing the stratification method in the sentence  "The first probabilistic sampling unit was the school and the second was the course to be chosen within each level". What are the levels/course? This is not addressed again in the article, adding further confusion.

Reply: We have improved the wording for this observation.

  • Given that the data is stratified, should the analyses have been conducted in a manner that integrates the complex survey structure?

Reply: One of the moderators considered was the School Vulnerability Index (SVI), which reflects to some degree the stratification of the sample. The strata were formed from the SVI and the type of dependency of each school. Due to this, we could not consider the variable used to stratify the sample as a control variable.

  • Although the sample is large, clarifying that it meets adequate statistical power in the participants section would be valuable.

Reply: Thanks for the comment. Due to the large sample size we do not take this consideration. Do you mean that we should calculate the statistical power for a punctual estimate? Or that we should calculate it for each of the estimated effect sizes?

  • In the instruments section, clarify to the reader that scale scores are being referred to, and not the scales themselves, when describing reliability. 

Reply: Thanks for this comment, we have corrected this point in the methodology.

  • Further justification of the adequacy of the PA measure in 2.3.2 is required given the modification of an existing measure's items without construct validity evidence presented.

Reply: It was explained why these items of the scale were considered and a CFA was carried out to analyze the construct validity.

  • In section 2.4, please clarify whether the PROCESS macro in SPSS centers the variables used to create the interaction terms. The potential for estimation bias due to strong collinearity between the base variables and their interaction terms is a core methodological consideration for moderation analyses.

Reply: To estimate the weight and significance of the interaction (moderation), PROCESS macro in SPSS is based on a regression analysis that considers the baseline variable, the moderating variable and the interaction between the two as independent variables. Since all these variables are included in the model, they exert control over each other when explaining the dependent variable. This comment is not entirely clear to us, we would be very grateful if you explain more what you mean by “whether the PROCESS macro in SPSS centers the variables used to create the interaction terms”.

  • Section 2.4 doesn't describe the analyses used for mean score differences presented in Table 2, nor does it explain why Age was split into arbitrary categories of 10-12 years old etc.

Reply: we have incorporated a description of the aforementioned analyzes and the organization of the age groups is explained within the same section. The latter deals with how children and adolescents are organized in the school system.

Results

  • In Table 2, statistical significance is presented as meaningful for mean score comparisons across the levels of the IVs, but realistically with a sample of ~9000 participants, even marginally-different comparisons will be significant. Present the effect sizes of these mean difference comparisons so the reader can understand practical significance. See the ASA's statement at https://www.amstat.org/asa/files/pdfs/p-valuestatement.pdf for further guidance on the use and interpretation of p

Reply: the effect sizes for each of the comparisons are now presented in the document and these sizes are incorporated in terms of their magnitude in the description of the results associated with Table 2.

  • Regression tables to present the results, instead of using statistical sentences in a conceptual figure of a moderation model, would be more appropriate when presenting the moderation findings. Currently information such as the intercept value are missing, as are estimates of effect size reported in-text such as an R2 value, and change in R2 value, following the introduction of the interaction component to understand the change in variance it is associated with. Statements such as "The difference in the variance of the relationship between PA and SWB is as high as 1332%." on page 7 are effectively meaningless without knowing what the R2 values are before/after introduction of the interaction term, as the reader is unable to judge the size of the effect prior to the introduction of the interaction term.

Reply: Thanks for the comment. The tables that were suggested were added and the mentioned data can be seen. The data of "1332%" was corrected, which actually referred to an increase of 33.33%.

  • In Figure 2, it is still unclear whether age has been entered as a continuous variable and the simple slope analysis is based on +- 1 SD as mentioned earlier in the results, or whether it has been evaluated as three nominal levels. Presenting a plot of the regression slopes varying by +- 1 SD and the mean, would be more effective compared to presenting a Figure containing statistical sentences here, and the former is more of the standard method of presenting moderation findings in sources such as Hayes' explanation of the PROCESS macro and how to employ it.

Reply: It was made explicit that for the moderation analysis, age was considered as a continuous variable, with the mean age level being the mean age of the sample and the low and high levels calculated from the addition / subtraction of a SD from the mean.

On this occasion, it was considered that presenting the tables and figures will convey the results of the investigation more clearly to the reader, instead of presenting a graph to compare slopes.

  • Section 3.2.2. would benefit from presenting a table of the regression results as noted earlier, in order to assist the reader with understanding how the model has been parameterised and the coefficients associated with each parameter.

Reply: Thank you for the comment, it was done this way.

  • Effect sizes are absent throughout the Results, which is a major limitation of this section. The reader is unable to judge the practical significance of any of the presented findings unfortunately.

Reply: For each of the statistically significant moderations, the increase in R2 was presented by adding the interaction to the regression analysis models, which serves to account for what percentage the moderations of the variance of the dependent variable explain. In addition, a relative comparison is made between the beta, for the different levels of the moderating variable.

Discusión

Starting the discussion, the authors note "In this sense, the aim of the present investigation was to analyze the moderating effect and possible interactions of the variables gender, age and socioeconomic level in the relationship between physical activity and subjective well-being in Chilean children and adolescents". Interactions between gender, age, and socioeconomic level were not clearly examined; the conceptual model in Figure 4 for example presents separate moderating effects of PA*Age and PA*Vulnerability, yet Table 3 presents PA*Age*Vulnerability findings. Reconciling the opening description of the study's aims against what has been presented in the Results, without the requested regression tables showing the coefficients of the model parameters, is challenging for the reader and reflects a major obstacle for clarity

Reply: we have improved the wording regarding the objectives of the study, seeking to improve consistency regarding the results presented.

  • The literature reviewed shortly after the previous point revisits the content from the Introduction, but requires more emphasis on describing how the current findings are similar/vary from previous research, without repeating the Introduction's content again.

Reply: we have substantially modified the discussion of the results in response to this and other comments regarding the same section.

  • Much of the literature described in the second paragraph of the section subtitled 'Moderating effect of gender' does not seem to have a clear bearing on the moderating effect of gender. It instead broadly describes PA and wellbeing for reasons that are not clear.

Reply: we have consolidated a discussion that improves the form and coherence of the writing around this observation.

  • A major limitation of the Discussion is tied to the coherency of the points being presented and the clarity of the writing. This section does not seem to have received the same care and editing as earlier sections, making it difficult for the reader. Consider the section 4.1 Strengths. Sentences feature missing words or content, such that by the time the reader reaches "Future research should consider these limitations..." (what limitations? Strengths are being described here?), it is clear that this section requires a significant revision for clarity.

Reply: we have addressed this observation trying to improve the coherence and wording of the discussion.

  • Consider the section 4.1 Strengths. Sentences feature missing words or content, such that by the time the reader reaches "Future research should consider these limitations..." (what limitations? Strengths are being described here?), it is clear that this section requires a significant revision for clarity.

Reply: we have corrected an error in the transcription to the document that generated this observation.

  • It is difficult in the conclusion for the reader to understand the importance of the study, and how its findings are novel, based on the preceding Discussion content. The authors should try to clearly articulate the novelty of their findings and how they differ from other similar works in the literature.

Reply: we hope, on the one hand, that the improvements made to the document in general will allow the study to be better articulated with the conclusions. In addition, we have improved the wording of this section based on this observation.

Reviewer 2 Report

Dear Authors,

It was a pleasure to revise the paper titled "Moderating role of Sociodemographic factors in the relationship between Physical activity and Subjective well-being in Chilean children and adolescents."
I consider it as important voice in the discussion on adolescents' well-being and its determinants. 
The article is methodologically well-prepared and well written. The only thing I hardly agree with is the statement in the introduction that only one paper from the cited review took into account the SES aspect. I myself had the opportunity to analyze this topic in the review of the review and, however, the theme is described much better in the literature than it is presented in the introduction to the article.
Here comes the review of the review I have mentioned above:

Kleszczewska D, Mazur J, Siedlecka J. Family, school and neighborhood factors moderating the relationship between physical activity and some aspects of mental health in adolescents. Int J Occup Med Environ Health. 2019 Jul 15;32(4):423-439. doi: 10.13075/ijomeh.1896.01389. Epub 2019 Jun 26. PMID: 31250837.

Author Response

October 10, 2021

Dear Reviewer

We welcome your comments regarding our study. We have considered your comments regarding the introduction in the improvements we made to the article. Additionally, we have incorporated into the study literature, the article suggested by you. We attach a new version that contains the improvements (highlighted in yellow) based on your observations and those of the second reviewer.

We hope on this occasion to adequately meet your expectations.

Sincerely,

Sergio Fuentealba Urra

Professor